# Bioinspired soft robots based on organic polymer-crystal hybrid materials with response to temperature and humidity

Xuesong Yang[1], Linfeng Lan[1], Xiuhong Pan[1], Qi Di[1], Xiaokong Liu [1],
Liang Li [2,3] ✉, Panče Naumov [2,4,5] ✉ & Hongyu Zhang [1] ✉

The capability of stimulated response by mechanical deformation to induce motion or actuation is the foundation of lightweight organic, dynamic materials for designing light and soft robots. Various biomimetic soft robots are constructed to demonstrate the vast versatility of responses and flexibility in shape-shifting. We now report that the integration of organic molecular crystals and polymers brings about synergistic improvement in the performance of both materials as a hybrid materials class, with the polymers adding hygroresponsive and thermally responsive functionalities to the crystals. The resulting hybrid dynamic elements respond within milliseconds, which represents several orders of magnitude of improvement in the time response relative to some other type of common actuators. Combining molecular crystals with polymers brings crystals as largely overlooked materials much closer to specific applications in soft (micro)robotics and related fields.

Mechanical softness is a common property of most biological tissues[1–3]. The softness of artificial materials that are intended to mimic these biogenic materials is germane to the emerging field of soft device engineering and, most prominently, for the emerging flexible soft electronics, where lightweight yet mechanically robust electronic devices can be bent, folded, twisted, compressed, stretched, or morphed into arbitrary shapes while their performance remains uncompromised[4–7]. The stiffness of small-molecule organic crystals (1–25 GPa) conveniently overlaps with some common soft materials ($10^{-4}$–25 GPa) while it also compares to those of soft biological tissues ($10^{-5}$–1 GPa)[8–10]. It is only recently that this unique standing of organic crystals in the materials global space has been recognized, triggering research efforts into an exploration of this class of soft materials as a foundation for bioinspired soft robotics[11–13]. These efforts in establishing organic crystals as a distinct class of engineering materials have opened prospects for their assessment against other, more common soft materials, with the organic crystals bringing an important added

value with a combination of long-range structural order and low density[14–20]. However, limitations such as slow response, fragility, and the uncontrollably small size of organic crystals have somewhat deterred engineers from immediately considering these materials as actuating elements. As one of the challenges, specific for the use in devices responsive to humidity, the crystal structures of common organic crystals are closely packed, ordered, and dense. Thus, they are impervious to water. An additional limitation has surfaced from systematic experimental and modeling studies, which have concluded that a natural upper limit binds the rate of deformation induced by chemical reactions in organic crystals[21–26]. The response times of crystals can be significantly improved by inducing mechanical instability by thermal stimulation (photothermal effect); however, this approach comes with significant energy losses due to thermal dissipation and very small, albeit rapid deformations[27–32]. These limitations with response time continue to pose challenges with other important functionalities related to deformation or actuation, such as

[1]State Key Laboratory of Supramolecular Structure and Materials, College of Chemistry, Jilin University, 130012 Changchun, P. R. China. [2]Smart Materials Lab, New York University Abu Dhabi, PO Box 129188 Abu Dhabi, UAE. [3]Department of Sciences and Engineering, Sorbonne University Abu Dhabi, PO Box 38044 Abu Dhabi, UAE. [4]Research Center for Environment and Materials, Macedonian Academy of Sciences and Arts, Bul. Krste Misirkov 2, MK–1000 Skopje, Macedonia. [5]Molecular Design Institute, Department of Chemistry, New York University, 100 Washington Square East, New York, NY 10003, USA. ✉e-mail: liang.li@sorbonne.ae; pance.naumov@nyu.edu; hongyuzhang@jlu.edu.cn

the recently widely explored optical signal transduction through organic crystals as waveguides and photonic circuits in the visible or near-infrared spectral regions[19,33–37]. Other materials such as polymers, elastomers, and liquid crystalline elastomers are often used to simulate biological behavior[38–42].

In this work, our approach to overcoming some of these challenges goes along with the strategy "the whole is greater than the sum of its parts" in the sense that combining materials of different natures could result in favorable properties that are not carried by either of the components alone. Specifically, here we capitalize on the fact that polymers span a much wider range of physical properties in the materials' property space relative to crystals. In contrast, organic crystals bear the advantage of long-range structural order. To that end, we combine the two classes of materials into a subfamily of hybrid, dynamically active elements using either casting of a single polymer or the layer-by-layer deposition of multiple polymers on the surface of molecular crystals. By selectively coating different faces of the crystals, which are inherently anisotropic bodies, this method provides access to micro- to centimeter anisotropic bilayer structures that are analogous to flexible bending beams. The dynamic elements are entirely organic, lightweight, elastic, and mechanically robust. Using combinatorial approaches and expanding the proposed strategy to deposit multiple polymer layers brings further about an array of functionalities that are not restrictive to the crystal's chemical composition or structure as long as the crystals are inherently flexible. These hybrid elements provide a paradigm shift in dynamic materials research, expand the scope of possible molecular crystal applications, and bring them closer to the requirements for their incorporation in soft robotic devices.

## Results and discussion

### Preparation of the organic polymer-crystal hybrid materials

The hygroresponsive mechanical functions of biological structures such as inflorescences, seeds, or tendrils can usually be approximated by simple deformations induced by mechanical instability that is caused by differential strain due to different tissues expanding or contracting at different rates. These mechanical deformations can ultimately be extrapolated to either simple bending or twisting. The bending is the simplest deformation where a bilayer changes its curvature due to a difference in the tensile forces between the layers. As only one of many documented examples[43–48] where this deformation leads to biological function, Fig. 1a shows a schematic illustration of the response of inflorescences of the rain lily (*Zephyranthes grandiflora*) to wetting[49]. The lily's flowers are open when wet, but they curl to close in dry air, thereby preventing the plant from dehydration. Although from the molecular engineering perspective, plants and organic crystals do not share any similarity, with the former being based on complex tissues while the latter are uniform in composition and are composed of molecules, on a macroscopic scale, mechanistically, they undergo similar and simple deformations that lead to visually resembling deformations[22,50]. In an effort to mimic some of these motions, we designed artificial dynamic elements that can bend upon external stimulation. The 'skeleton' of the active elements are centimeter-long slender elastic crystals of the three organic compounds 1–3 shown in Fig. 1b, c. These compounds were selected by screening the literature for reported elastic crystals (Supplementary Table 1, Supplementary Figs. 1–3)[51–53]. Single crystals of compounds 1–3 are indeed elastic, and they bend reversibly upon

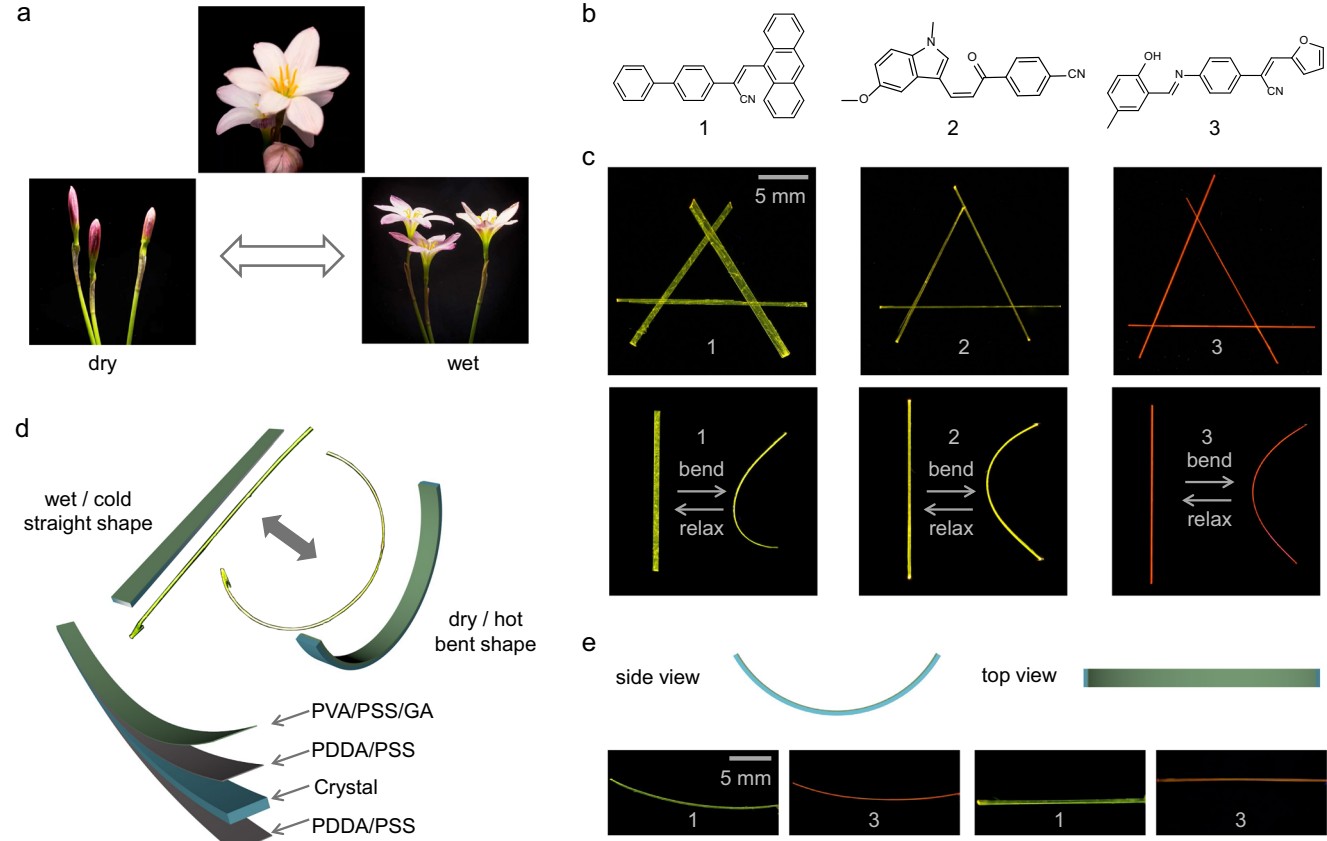

**Fig. 1 | Concept and preparation of hybrid flexible crystals. a** Photographs of the rain lily (*Zephyranthes grandiflora*) and its response to dry/wet conditions. **b** Chemical structures of the elastic crystals 1–3. **c** Photographs of straight and bent crystals of 1–3 recorded under UV light for contrast. **d** Schematic of the layered structure of the organic polymer-crystal hybrid materials P²//1–3. Bending occurs

when the organic polymer-crystal hybrid materials are placed in a dry or hot environment due to the differential strain that develops at the phase boundaries. **e** Schematic and optical images of the side and top view of bent P²//1,3 crystals recorded with UV light radiation (RH = 40.1%).

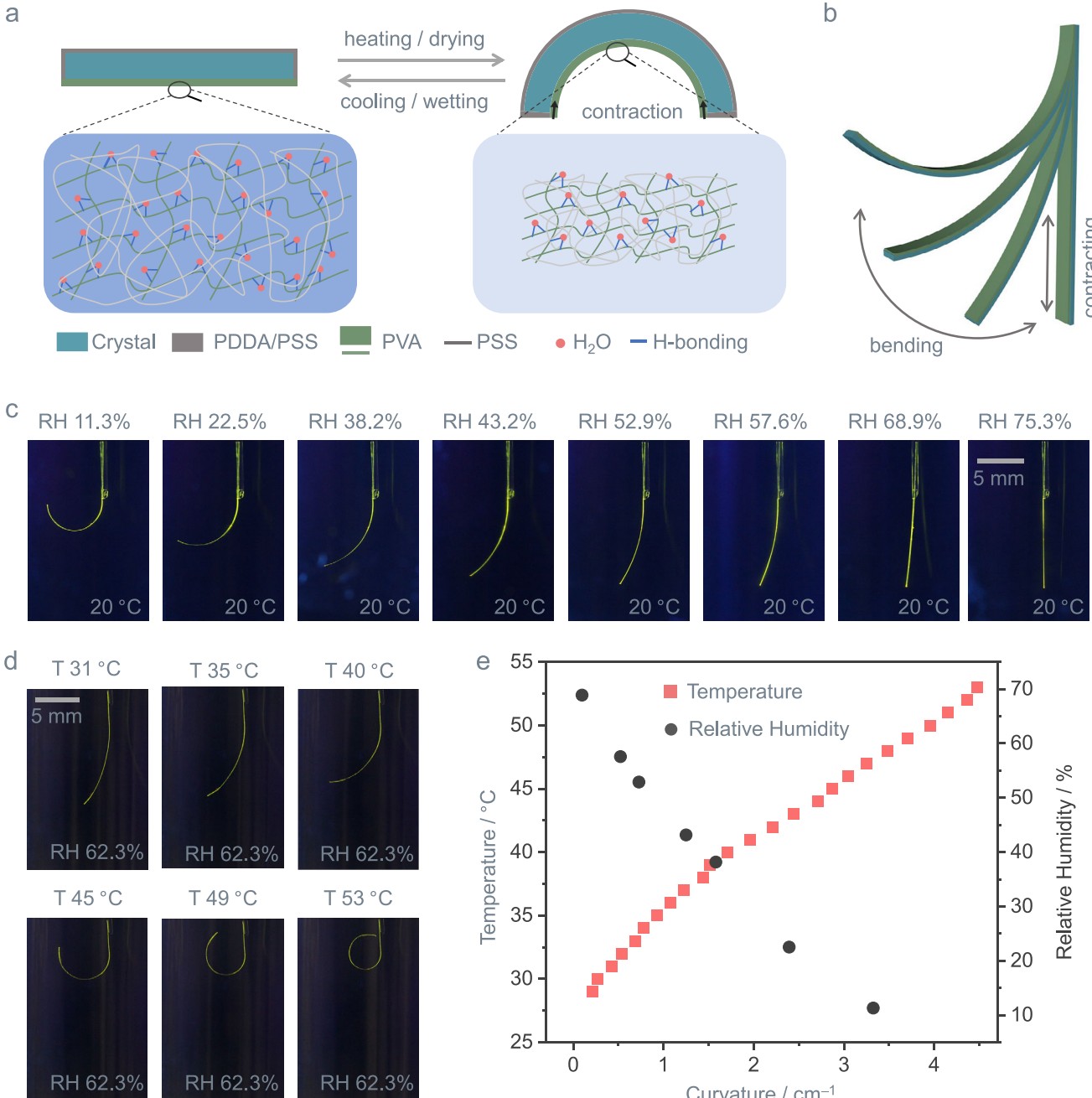

**Fig. 2 | Humidity and temperature sensing capability of organic polymer-crystal hybrid materials, P²//1. a** Schematic demonstration of the design principle of bendable hybrid crystalline actuators. **b** Principles of bending of the organic polymer-crystal hybrid materials. **c** Photographs of bent P²//1 induced by relative humidity (RH) changes at the temperature of 20 °C. **d** Photographs of bent P²//1 at different temperatures with a consistent RH value of 62.3%. **e** Bending curvature of P²//1 plotted as a function of temperature and relative humidity.

the application of lateral force. At the same time, they are also conveniently fluorescent and can be easily visualized under UV light against a dark background.

The long crystals were first coated on all faces with a mixture of poly(diallyldimethylammonium chloride) (PDDA) and poly(sodium 4-styrenesulfonate) (PSS) as an approximately $300 \pm 100$-nm-thick layer (Fig. 1d; Supplementary Fig. 4, Supplementary Methods). Subsequently, a $700 \pm 100$-nm-thick layer of a mixture of polyvinyl alcohol (PVA), poly(sodium 4-styrenesulfonate) (PSS), and glutaraldehyde (GA) was deposited on one of the crystal's wide faces (Supplementary Fig. 5), resulting in hybrid structures PVA/PSS/GA//PDDA/PSS//1–3 that for convenience hereafter are referred to as P²//1–3. The mechanical properties of the organic polymer-crystal

hybrid materials were investigated at different conditions by three-point bending tests (Supplementary Fig. 6). PVA is a common hygroscopic polymer with low critical solubility temperature (LCST) that is well-known to undergo reversible swelling by the formation of hydrogen bonds[54]. GA was used as a cross-linking agent that enables PVA to form a three-dimensional network structure. The sensitivity of PVA to humidity and temperature is further enhanced by PSS[55]. As shown in Fig. 1e, the coated crystals retain their elasticity and can be bent in low relative humidity, RH = 40.1%. Since only one of the two wide faces of the crystal was coated with polymers, the polymer shrinks, resulting in a differential strain that translates into a bending moment. This is observed as the macroscopic bending of the hybrid crystal.

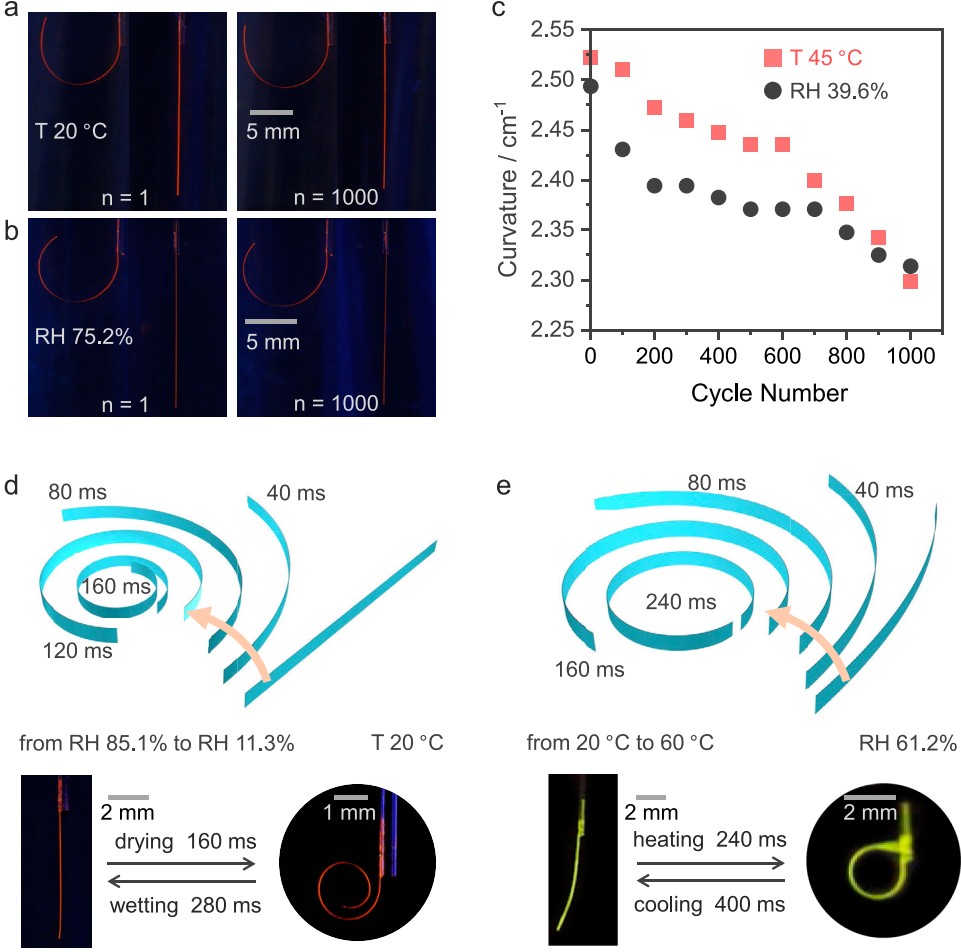

**Fig. 3 | Actuation cyclability and durability of the organic polymer-crystal hybrid materials. a, b** Photographs of P²//3 of straight and bent shapes after cycle 1 and after cycle 1000, induced by humidity (**a**) and temperature (**b**), respectively, under UV light for enhanced contrast against the background. **c** Curvature change of P²//3 during cyclic operation upon variation of either temperature or humidity. **d, e** Rate of response of P²//3 to humidity (**d**) and P²//2 to temperature changes (**e**) and images of the actual crystals in straight and bent states.

## Response of organic polymer-crystal hybrid materials to humidity and temperature

The organic polymer-crystal hybrid materials were composed of two parts. PDDA/PSS//1–3 was utilized as a substrate for the hybrid materials, which can bend under external forces due to a combination of intermolecular interactions and, most notably, by capitalizing on the strength of hydrogen bonds (Fig. 1c)[53,56,57]. The mixture of PDDA and PSS was used to improve the adhesion of the PVA/PSS/GA layer onto the crystal surface[58,59]. PVA/PSS/GA layer functions as the driving element. As the temperature increases or the humidity decreases, the water molecules inside the polymer are released, causing shrinking. This change in length ultimately causes the material to bend. Conversely, when the temperature is decreased, or the humidity is increased, the polymer hydroxyl groups form hydrogen bonds with the absorbed water molecules. This results in expansion, and the shape is restored (Fig. 2a), as qualitatively supported by the vibrational spectra (Supplementary Fig. 7)[54,60–62]. It is worth mentioning that, similar to what has been pointed out before[51,63–65], the approach described here is not limited to the specific organic crystals or polymers that were used; in principle, any elastic crystal could be combined with a hygroresponsive polymer, provided that the crystal is flexible and the polymer solution can be deposited and adheres firmly onto the crystal surface without dissolving it. The strategy therefore provides a wealth of hybrid soft materials.

By incorporating hygroresponsive and thermally responsive polymers, the combination of PVA, PSS, and GA was anticipated to respond to changes in both humidity and temperature (Fig. 2a, b). Indeed, placing P²//1 that has been affixed to a solid support at one end in an atmosphere with controlled RH between 11.3 and 75.3% for 20 min induces visible deformation to a bent structure which maintains its curvature if the humidity level is kept constant (Fig. 2c). Being anisotropic and composed of three components of different thermal expansion with clear phase boundaries (Supplementary Fig. 4), the hybrid elements were also expected to deform when subjected to temperature change. In line with this expectation, temperature variation from 20 to 60 °C results in strong and visible bending (Fig. 2d, Supplementary Fig. 8). As shown in Fig. 2e, the bending curvature of the hybrid elements increases when humidity increases, or the temperature decreases. This humidity- and temperature-dependent deformation provides an opportunity to use, at least in principle, any elastic organic crystal coated with a polymer as an actuator, even though single crystals of closely packed, homogeneous molecular structures are rarely responsive to humidity and temperature changes.

## Durability and sensitivity of organic polymer-crystal hybrid materials

Both thermally and humidity-induced deformations occur due to differential strain that develops and favor these elements as actuators with a dual response. Within a more general context, high mechanical robustness, cyclability in operation over prolonged usage, and high response sensitivity stand as some of the main prerequisites for

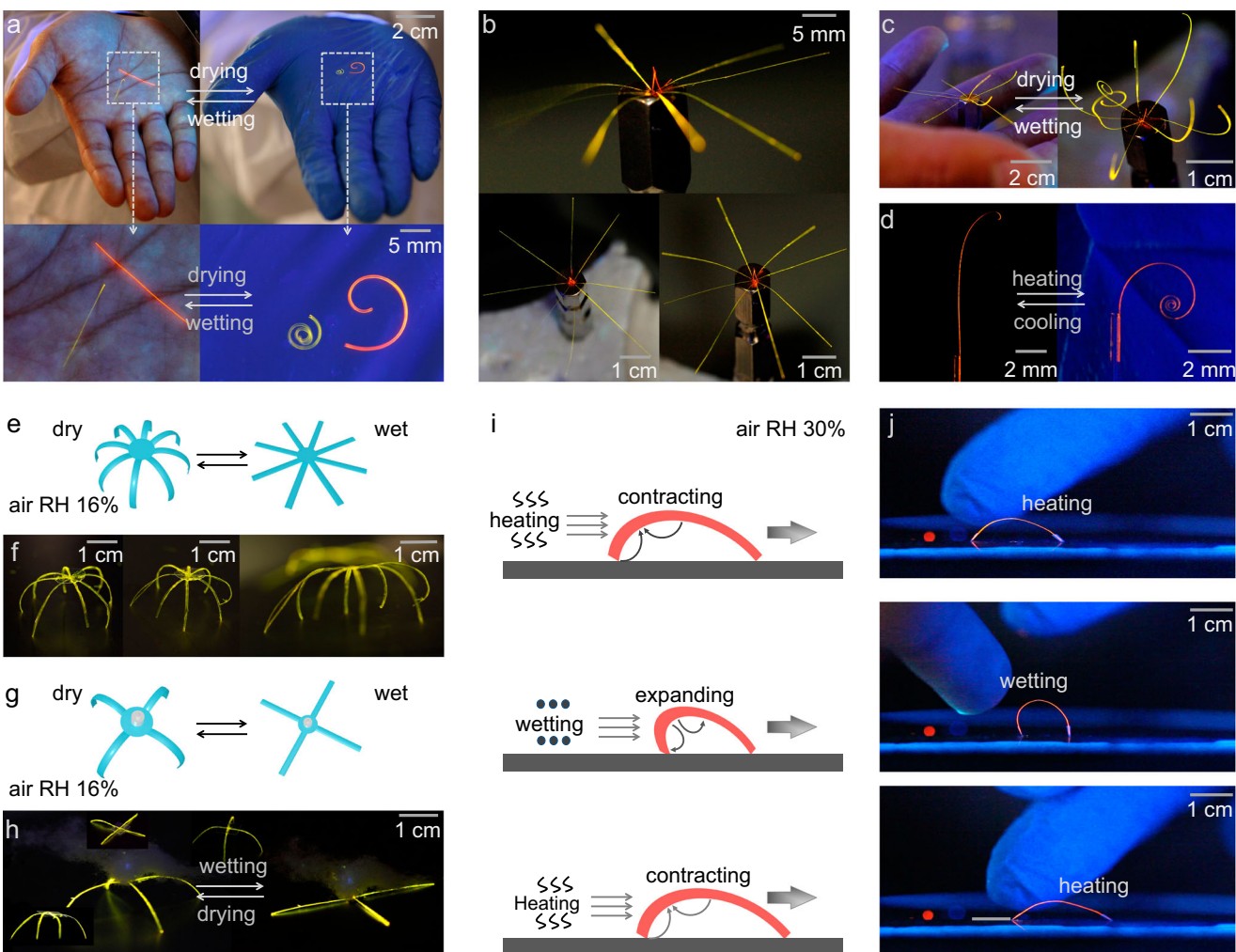

**Fig. 4 | Soft robots based on hybrid polymer-crystal materials. a** Crystals of P²//2,3 straighten when they are placed on a bare palm and bend when they are placed on a nitrile glove due to response to the variation in humidity. **b** A model "inflorescence" made of PDDA/PSS//2 and crystal 3. **c** An artificial model "inflorescence" of rain lily (*Z. grandiflora*) prepared from P²//2 and crystal 3. **d** A model of a plant tendril made of P²//3 which can curl by a change in temperature (Supplementary Movie 3). **e, f** Schematic diagram (**e**) and photographs (**f**) of a soft robot capable of performing a spider-like motion (Supplementary Movie 4). **g, h** Schematic diagram (**g**) and photographs (**h**) of a soft gripper made of P²//1 (Supplementary Movie 5). **i** Schematic representation of the mechanism of 'walking' of organic polymer-crystal hybrid materials across a surface induced by periodic changes in aerial humidity. **j** Snapshots of the 'walking' of a hybrid crystal of P²//3 (Supplementary Movie 6).

applying any material as an actuator in reconfigurable devices such as soft robots. Here, the cyclability of the hybrid crystals was tested with P²//3 as an example. At 20 °C, alternating exposure of P²//3 to RH levels of 85.1 and 39.6% did not show any significant variation in its bending capability even after 1000 cycles (Fig. 3a, Supplementary Fig. 9). The hybrid elements are robust even with thermal stimulation; at a constant humidity level of 75.2% RH, P²//3 retains its bending ability when it is alternately heated up to 45 °C and cooled to 20 °C up to at least 1000 times (Fig. 3b). After 1000 cycles, the bending curvature decreases by 8.7% by thermal cycling and 7.2% by cycling in humidity (Fig. 3c, Supplementary Fig. 10). The rate of response of the hybrid elements P²//2,3 was also studied. Immediate humidity increase from 11.3 to 85.1% results in a response of P²//3 by straightening within 280 ms. As shown in Fig. 3d and Supplementary Movie 1, upon quick drying of the surrounding air from 85.6 to 11.3% RH, the actuator responds within 160 ms. As further demonstrated in Fig. 3e and Supplementary Movie 2, the immediate temperature change from 20 to 60 °C results in bending of P²//2 within 240 ms, and cooling recovers the straight shape in 400 ms. Notably, the kinetics of the bending response of these hybrid crystals to changes in humidity and temperature is significantly improved when compared

to the response of similar materials to humidity (1–2 min)[63] and light (20 s)[66].

## Application of organic polymer-crystal hybrid materials as soft robots

The core of the operational principles of robots is the simple mechanical reconfiguration of materials or active devices, such that eventually result in capability to move in space and to perform other tasks, such as gripping or release of objects. The generality of our approach in this work to investigate the response to humidity and temperature prompted us to explore the possibility of utilizing the hybrid elements as dynamic components of simple soft robots. The inspiration comes from the fact that the strong and reversible deformations of the hybrid elements, such as bending, could change the shape of the elements and translate momentum onto another object or drive the process of relocation of that object in space. The high sensitivity to changes in temperature and humidity was therefore utilized to demonstrate the response of the hybrid elements that may inspire the construction of actual prototypical soft robots.

As shown in Fig. 4a, a single hybrid element held by a needle remains straight while approached with a bare palm (high temperature,

high humidity), but it bends when it is approached with a hand in a nitrile glove (high temperature, low humidity). Breathing toward the sample or approaching it with a bare palm represents a simple way to change the environment of the sample. In Fig. 4b, the ends of several PDDA/PSS//2 crystals were first glued together into the shape of a flower petal, and the stamen was decorated by crystal 3. As illustrated in Fig. 4c, PVA/PSS/GA was deposited on the top side of PDDA/PSS//2. Due to the low humidity (RH = 21%), multiple $P^2$//2 elements bend simultaneously. These artificial petals open up when approached by a bare palm (high humidity, high temperature), much like the inflorescence of a rain lily (Fig. 4c). As shown in Fig. 4d, to further explore the biomimetic functions of the material, we tried to mimic a single tendril of a plant, where the element responds when approached with a hand in nitrile glove (Supplementary Movie 3)[67]. Building upon these exemplary dynamic operations, a collective hybrid element was constructed by gluing together multiple hybrid crystals at one end onto a polyethylene terephthalate (PET) plate placed above a smooth silicon wafer (Fig. 4e, f, Supplementary Movie 4). Breathing toward this soft robotic structure increased the local humidity and induced concomitant bending of the individual elements, the deformation visually resembling motion of spider's legs. We also showed that this soft robot could operate as a miniature gripping device; with a body weight of 15 mg, it can lift an object that weighs 300 mg, that is, 20 times its weight (Fig. 4g, h, Supplementary Movie 5)[68,69].

The dynamic functionality of the hybrid materials was further explored to build simple walking robots. As shown in Fig. 4i, j and Supplementary Movie 6, the actuator is capable of walking on a smooth silicon wafer when it is alternatively exposed to heat and humidity. Initially, the inner layer shrinks upon heating, and the structure bends downward, forming an arc. As the humidity increases, the inner layer expands, and the structure stretches. Due to imperfections in shape, the robot 'walks' sidewise by making individual steps, driven by the interfacial strain between the inner PVA/PSS/GA layer and the crystal. We confirmed that this actuator was able to move over a distance of up to 3 cm within 19 s, although the rate depends on the external gradient in heat and humidity. The relevance of the deformation and motion of these hybrid materials goes beyond their visual appeal and provides evidence that they can be used as sensitive, responsive actuation elements that are driven both by variations in temperature and humidity.

In summary, we report a family of hybrid polymer-crystal materials that undergo deformations such as bending driven by temporal and/or spatial changes in humidity and temperature. The hybrid materials exhibit fast bidirectional bending facilitated by both temperature and humidity. A favorable combination of the properties of polymers and flexible organic crystals was accomplished by depositing the polymers onto the surface of elastic organic crystals. By exploiting the response of these hybrid materials to humidity and temperature in various geometries and combinations, we successfully simulated the movements and deformations of various living organisms in nature, such as those of certain inflorescences and plant tendrils. The joint action of temperature and humidity expands the palette of stimuli-induced deformation modes and kinetics of flexible organic crystals and opens up prospects for more direct application of flexible organic crystals in flexible devices, soft robotics, and bionics.

## Methods

### Materials
The solvents and starting materials used in the syntheses were obtained from commercial sources and used without further purification. The compounds 1–3 were synthesized following established procedures (for details, see Supplementary Figs. 1–3 and 11–16)[51–53]. To prepare the samples, dichloromethane solutions of compounds 1–3 were placed in test tubes. Approximately triple volume of ethanol was then added along the walls of the tube without disturbing the surface

of the solution, and the solutions were left for diffusion to occur. Needle-shaped crystals of compounds 1–3 were obtained after 1–2 weeks at room temperature.

### Fabrication of organic polymer-crystal hybrid materials $P^2$//1–3
The long crystals were immersed in 1 mg mL$^{-1}$ solution of poly(diallyldimethylammonium chloride) (PDDA) for 20 min, followed by a 1-min rinse with distilled water. Then, the crystals were immersed in 1 mg mL$^{-1}$ solution of poly(sodium 4-styrenesulfonate) (PSS) for 20 min and rinsed with distilled water for 1 min. The coated crystals were obtained by repeating the above steps. By using a needle tip, a mixture of polyvinyl alcohol (PVA), poly(sodium 4-styrenesulfonate) (PSS), and glutaraldehyde (GA) was deposited on one of the crystal's wide faces. As the solvent evaporated at room temperature, a polymer film formed on the surface of the crystal, and $P^2$//1–3 were obtained.

## Data availability
All data are available from the corresponding authors upon request.

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

## Acknowledgements
This work was supported by the National Natural Science Foundation of China (52173164) and a fund from New York University Abu Dhabi. This material is based upon works supported by Tamkeen under NYUAD RRC Grant No. CG011.

## Author contributions
X.Y., L.La., X.P., Q.D., and X.L. performed the experiments. L.Li., P.N., and H.Z. supervised the experiments. H.Z. and P. N. conceived the project.

## Competing interests
The authors declare no competing interests.
