## [Peer Review File · Nature Communications]

Bioinspired Soft Robots based on Organic Polymer-Crystal Hybrid Materials with Response to Temperature and HumidityReviewers' Comments:

Reviewer #1:

Remarks to the Author:

In this paper, the authors combine the mechanical properties of polymers with those of organic crystals and demonstrate the shape-changing properties of the resulting hybrid organic crystals. This is of fundamental interest in terms of control over the responsiveness of materials, but also in terms of their usefulness: it is known that coatings such as paints (coalesced latex beads) or cellulose-based, sustainable coatings would benefit from strategies to encode a priori contradictory mechanical properties in the same materials. For example, it is often necessary to combine ductility and robustness. Here, Zhang et al. have designed and engineered materials that combine flexibility and robustness with also dual-responsiveness, and this adds to the toolbox of strategies for the design and development of thin responsive films, and maybe also soft robotics.

The experimental work is described appropriately. All experiments are extensively documented. I have minor comments and suggestions:

- In Figure 1a and throughout the text, the authors compare their system to the rain lily. This is appealing and enjoyable to read, but in terms of molecular engineering, do a plant and an organic crystal share many design or structural similarities? Perhaps the authors could add a couple of sentences to clarify this.

- The manuscript contains extensive information regarding the actuating properties of these hybrid flexible crystals, which is great. Some insight into the mechanical properties of these materials would have also been helpful, as it would have provided a better understanding of where the performances originate. For example, the manuscript mentions "elastic materials," but can their elasticity be quantified? What about the Young modulus of these materials? Is it modified by irradiation, or by the presence of humidity?

- Other authors have developed actuating materials, with analogies to the plant or animal world, and it might be nice to cite some of these works for the benefit of contextualization to the broad readership - even if these examples include only polymers in their design: Priimagi, Schenning, Aizenberg, Katsonis, Studart etc.

Reviewer #2:

Remarks to the Author:

Comments to the authors - "Bioinspired Soft Robots based on Organic Polymer-Crystal Hybrid Materials"

In this work, the authors explored the utility of stimuli-responsive hybrid materials to demonstrate the bio-inspired mechanical response of hybrid organic crystalline-polymer materials for soft robotics and actuation. This manuscript presents the use of polymer-coated crystals to mimic the mechanical response of lily flowers in dry/wet conditions using temperature and relative humidity. This work is novel in design strategy and presents an application for soft robotics. Therefore, I recommend the consideration of this manuscript for publication in Nature Communications.

The following concerns should be carefully addressed before considering this manuscript for publication in any journal.

1. The authors write, "Both thermally and humidity-induced deformations occur as a result of the differential strain that develops and favor these elements as actuators with a dual response."

However, given the experimental context, the external stimuli are isotopically distributed around the hybrid material. What really triggers the differential strain caused by the system?

2. It is better to replace "hybrid organic crystals" with "Organic Polymer-Crystal Hybrid Materials."

3. Page 2, line 63, I suggest the modification of the sentence "...optical signal transduction through

organic crystals as waveguides" to "explored optical signal transduction through organic crystals as waveguides and photonic circuits" – ref. Chem. Commun. (2022), 58, 3415-3428.

4. The following sentence needs to be modified.

"At the core of the operational principles of robots are simple reconfigurations of materials or active devices such as reshaping and motility that eventually result in transportation in space and other tasks such as, for instance, gripping or release of objects."

Reviewer #3:

Remarks to the Author:

Opinion on the Nature Com. ms. entitled: "Bioinspired Soft Robots based on Organic Polymer-Crystal Hybrid Materials with Response to Temperature and Humidity" by Xuesong Yang, Linfeng Lan, Xiuhong Pan, Qi Di, Xiaokong Liu, Liang Li*, Panče Naumov*, and Hongyu Zhang.

This is a very interesting ms. on a family of hybrid organic crystalline-polymer materials that undergo deformations such as bending that are driven by gradients in humidity and temperature. The ms. contains description of synthesis and macroscopic properties of these materials which are combination of polymers and flexible organic crystals. The hybrid materials made respond very quickly (within milliseconds). The whole concept is very interesting and the properties of such hybrid materials are both well described and also illustrated in the form of short videos (which is very convincing at least in my case).

In my opinion this work is relevant for the field and also potentially could be relevant for some related fields such as artificial molecular machines.

Utilising the response of these hybrid organic crystal to humidity and temperature, the PT Authors successfully simulated movements and deformation patterns of various organisms such as inflorescences and plant tendrils. Such materials open new prospects for some interesting new application of flexible organic crystals in different flexible devices, soft robotics and bionics.

I must say that I see only one downside to this ms. which is the lack of interpretation of the macroscopic behaviour of these hybrid materials at the microscopic level. As a reviewer, I would like to know the molecular/atomic mechanism of action of these hybrid materials. In my opinion, knowledge of this mechanism at the molecular/atomic level will allow for optimization of the properties of such materials and also should give a chance to make some new classes of artificial nanodevices which could also do some useful work at the nanolevel of organisation of matter (although the macroscopic behaviour of these materials is also very interesting and important).

Conclusion: I am for publication of this article when some more information about the molecular/atomic mechanism of action of these materials is added.

Response to reviewers' comments on the manuscript "Bioinspired Soft Robots based on Organic Polymer-Crystal Hybrid Materials with Response to Temperature and Humidity"

Response to the comments from Reviewer #1:

Overall Comment: *In this paper, the authors combine the mechanical properties of polymers with those of organic crystals and demonstrate the shape-changing properties of the resulting hybrid organic crystals. This is of fundamental interest in terms of control over the responsiveness of materials, but also in terms of their usefulness: it is known that coatings such as paints (coalesced latex beads) or cellulose-based, sustainable coatings would benefit from strategies to encode a priori contradictory mechanical properties in the same materials. For example, it is often necessary to combine ductility and robustness. Here, Zhang et al. have designed and engineered materials that combine flexibility and robustness with also dual-responsiveness, and this adds to the toolbox of strategies for the design and development of thin responsive films, and maybe also soft robotics.*

The experimental work is described appropriately. All experiments are extensively documented. I have minor comments and suggestions:

Response: We thank the Reviewer for their valuable comments, and for recognizing the significance of the work presented in our manuscript. A point-by-point response to the Reviewer's comments are provided below.

Comment: 1. - *In Figure 1a and throughout the text, the authors compare their system to the rain lily. This is appealing and enjoyable to read, but in terms of molecular engineering, do a plant and an organic crystal share many design or structural similarities? Perhaps the authors could add a couple of sentences to clarify this.*

Response: We thank the Reviewer for pointing this out, and we concur with the Reviewer that the analogy of the (homogeneous) material with a complex biological system, may a little too distant in view of their structural mechanisms that lead to dynamic response. By making this analogy, we were hoping to highlight that at a macroscopic level, the motility of plants can be approximated to simple motions such as bending or twisting. In that regard, the recently researched mechanical responses of organic crystals are analogous in the sense that they can also be reduced to simple mechanical deformations, such as bending and twisting. Although clearly they are governed by different mechanisms, within the classical elasticity theory these systems have many similarities in terms on the main forces and strains that develop, and the consequences of them on the macroscopic shape. In the living world, many of these motions that are ultimately aimed to provide means for reproduction and dispersal, are driven by light, heat and even changes in humidity. In the crystals, on the other hand, these changes are governed by changes at a molecular scale and their amplification to a macroscopic scale via intermolecular interactions. In the work presented in our manuscript, the response of inflorescences of the rain lily (*Zephyranthes grandiflora*) to contact with water was compared to the behavior of the response of the organic crystals to humidity, because in both case these phenomena are rooted in the development of differential strain by non-uniform absorption of water on opposite sides of the mechanically active elements. In an attempt to clarify this pointm the following text has been added on page 3 in the revised manuscript:

Added text: “Although from the molecular engineering perspective, plants and organic crystals do not share any similarity with the former being based on complex tissues while the latter are uniform in composition and are composed of molecules, on a macroscopic scale mechanistically they undergo similar and simple deformations that leads to visually resembling deformations.^{22,55}”

Comment: 2. - *The manuscript contains extensive information regarding the actuating properties of these hybrid flexible crystals, which is great. Some insight into the mechanical properties of these materials would have also been helpful, as it would have provided a better understanding of where the performances originate. For example, the manuscript mentions "elastic materials," but can their elasticity be quantified? What about the Young modulus of these materials? Is it modified by irradiation, or by the presence of humidity?*

Response: We thank the Reviewer for bringing this important point to our attention. In the revised version, we have added the analysis of the mechanical properties of the hybrid flexible crystals at different conditions. This includes three-point bending experiments and the corresponding results. Moreover, a stain-stress diagram and the results from the measurements of the Young's modulus of the material have been added to the revised version of the Supporting Information. The following text has been added on page 3 in the revised manuscript:

Added text: “The mechanical properties of the organic polymer-crystal hybrid materials were investigated at different conditions by three-point bending tests (Supplementary Figure 3).”

New Supplementary Figure 3. Stress-strain profiles of the hybrid materials under different conditions. (a) Crystal 1 at ambient condition. (b) P²//1 at ambient condition. (c) P²//1 under UV light irradiation. (d) P²//1 tested in a stream of air at high humidity.

Comment: 3. - Other authors have developed actuating materials, with analogies to the plant or animal world, and it might be nice to cite some of these works for the benefit of contextualization to the broad readership - even if these examples include only polymers in their design: Priimagi, Schenning, Aizenberg, Katsonis, Studart etc.

Response: This is an important point, and we thank the Reviewer for bringing it up. We are familiar with the excellent work of these authors who work mainly on analogous phenomena in polymers. The examples of actuating materials developed by other authors have been cited in the text, and they are included as references 39-43.

The following text has been added on page 2, line 66 in the revised manuscript:

Added text: "Other materials such as polymers, elastomers and liquid crystalline elastomers are often used to simulate biological behavior.³⁹⁻⁴³"

Response to the comments from Reviewer #2:

Overall Comment: *In this work, the authors explored the utility of stimuli-responsive hybrid materials to demonstrate the bio-inspired mechanical response of hybrid organic crystalline-polymer materials for soft robotics and actuation. This manuscript presents the use of polymer-coated crystals to mimic the mechanical response of lily flowers in dry/wet conditions using temperature and relative humidity. This work is novel in design strategy and presents an application for soft robotics. Therefore, I recommend the consideration of this manuscript for publication in Nature Communications.*

The following concerns should be carefully addressed before considering this manuscript for publication in any journal.

Response: We are very grateful to the Reviewer for their constructive and encouraging comments. We provide below a point-by-point response to their comments and suggestions.

Comment: 1. *The authors write, “Both thermally and humidity-induced deformations occur as a result of the differential strain that develops and favor these elements as actuators with a dual response.” However, given the experimental context, the external stimuli are isotopically distributed around the hybrid material. What really triggers the differential strain caused by the system?*

Response: We thank the Reviewer for the careful reading. We would like to note that the polymer is only deposited on one of the crystal's wide faces (page 3, line 104). When humidity or temperature changes, the deposited polymer shrinks, and since it is intimately bound to the crystal, this causes interfacial strain on that face. The strain translates into a bending moment, and as a result, the hybrid crystals bends. This is not possible with the pure crystals, where all faces are uniformly affected by humidity. To clarify this point, the following texts has been revised (page 4, line 114):

Original text: “As shown in Figure 1e, the coated crystals retain their elasticity and can be bent in low humidity (RH = 40.1%).”

Revised text: “As shown in Figure 1e, the coated crystals retain their elasticity and can be bent in low humidity (RH = 40.1%). Since only one of the two wide faces of the crystal was coated with polymer, the polymer shrinks, resulting in differential strain that translates into a bending moment. This is observed as macroscopic bending of the hybrid crystal.”

Comment: 2. It is better to replace “hybrid organic crystals” with “Organic Polymer-Crystal Hybrid Materials.”

Response: We thank the reviewers for their suggestions. “Hybrid organic crystals” has been replaced with “organic polymer-crystal hybrid materials” throughout the text. The following sections have been revised accordingly:

Original text: “(d) Schematic of the layered structure of the hybrid organic crystals P²//1–3. Bending occurs when the hybrid organic crystals are placed in dry or hot environment due to the differential strain that develops at the phase boundaries.”

Revised text: “(d) Schematic of the layered structure of the organic polymer-crystal hybrid materials P²//1–3. Bending occurs when the organic polymer-crystal hybrid materials are placed

in dry or hot environment due to the differential strain that develops at the phase boundaries.”

Original text: “Figure 2. Humidity and temperature sensing capability of hybrid organic crystals, P²//1. (a) Schematic demonstration of the design principle of bendable hybrid crystalline actuators. (b) Principles of bending of the hybrid organic crystals.”

Revised text: “Figure 2. Humidity and temperature sensing capability of organic polymer-crystal hybrid materials, P²//1. (a) Schematic demonstration of the design principle of bendable hybrid crystalline actuators. (b) Principles of bending of the organic polymer-crystal hybrid materials.”

Original text: “Figure 3. Actuation cyclability and durability of the hybrid organic crystals.”

Revised text: “Figure 3. Actuation cyclability and durability of the organic polymer-crystal hybrid materials.”

Original text: “(i) Schematic representation of the mechanism of ‘walking’ of a hybrid organic crystals across a surface induced by periodic change in aerial humidity.”

Revised text: “(i) Schematic representation of the mechanism of ‘walking’ of an organic polymer-crystal hybrid materials across a surface induced by periodic change in aerial humidity.”

Original text: “

Supplementary Video 1. Bending of the hybrid organic crystals induced by humidity.

Supplementary Video 2. Bending of the hybrid organic crystals induced by temperature.

Supplementary Video 4. Simulation of a spider-like motion of using hybrid organic crystals.

Supplementary Video 5. Gripping device based of hybrid organic crystals.

Supplementary Video 6. Walking of the hybrid organic crystals across a surface.”

Revised text: “

Supplementary Video 1. Bending of the organic polymer-crystal hybrid materials induced by humidity.

Supplementary Video 2. Bending of the organic polymer-crystal hybrid materials induced by temperature.

Supplementary Video 4. Simulation of a spider-like motion of using organic polymer-crystal hybrid materials.

Supplementary Video 5. Gripping device based of organic polymer-crystal hybrid materials.

Supplementary Video 6. Walking of the organic polymer-crystal hybrid materials across a surface.”

Comment: 3. *Page 2, line 63, I suggest the modification of the sentence “...optical signal transduction through organic crystals as waveguides” to “explored optical signal transduction through organic crystals as waveguides and photonic circuits” – ref. Chem. Commun. (2022), 58, 3415-3428.*

Response: This is a very important point, and we thank the Reviewer for bringing it to our attention. The sentence “...optical signal transduction through organic crystals as waveguides” has been modified to “explored optical signal transduction through organic crystals as waveguides and photonic circuits”. The following changes have been made in the main text (page 2, line 61) and the relevant reference has been included in the reference list:

Original text: “These limitations with response time continue to pose challenges with some other

important functionalities that are related to deformation or actuation, such as the recently widely explored optical signal transduction through organic crystals as waveguides in the visible or near-infrared spectral regions. ^{33–37}”

Revised text: “These limitations with response time continue to pose challenges with some other important functionalities that are related to deformation or actuation, such as the recently widely explored optical signal transduction through organic crystals as waveguides and photonic circuits in the visible or near-infrared spectral regions. ^{33–38}”

Comment:4. *The following sentence needs to be modified. “At the core of the operational principles of robots are simple reconfigurations of materials or active devices such as reshaping and motility that eventually result in transportation in space and other tasks such as, for instance, gripping or release of objects.”*

Response: We thank the Reviewer for bringing this to our attention. Accordingly, the following sentence has been modified (page 7, line 265):

Original text: “At the core of the operational principles of robots are simple reconfigurations of materials or active devices such as reshaping and motility that eventually result in transportation in space and other tasks such as, for instance, gripping or release of objects.”

Revised text: “The core of the operational principles of robots are simple reconfigurations of materials or active devices such as reshaping and motility that eventually result in transportation in space and other tasks such as gripping or release of objects.”

Response to the comments from Reviewer #3:

Overall Comment: *This is a very interesting ms. on a family of hybrid organic crystalline-polymer materials that undergo deformations such as bending that are driven by gradients in humidity and temperature. The ms. contains description of synthesis and macroscopic properties of these materials which are combination of polymers and flexible organic crystals. The hybrid materials made respond very quickly (within milliseconds). The whole concept is very interesting and the properties of such hybrid materials are both well described and also illustrated in the form of short videos (which is very convincing at least in my case).*

In my opinion this work is relevant for the field and also potentially could be relevant for some related fields such as artificial molecular machines.

Utilising the response of these hybrid organic crystal to humidity and temperature, the PT Authors successfully simulated movements and deformation patterns of various organisms such as inflorescences and plant tendrils. Such materials open new prospects for some interesting new application of flexible organic crystals in different flexible devices, soft robotics and bionics. I must say that I see only one downside to this ms. which is the lack of interpretation of the macroscopic behaviour of these hybrid materials at the microscopic level. As a reviewer, I would like to know the molecular/atomic mechanism of action of these hybrid materials. In my opinion, knowledge of this mechanism at the molecular/atomic level will allow for optimization of the properties of such materials and also should give a chance to make some new classes of artificial nanodevices which could also do some useful work at the nanolevel of organisation of matter (although the macroscopic behaviour of these materials is also very interesting and important).

Response: We are very thankful to the Reviewer, who apparently is a very knowledgeable expert in this field, for recognizing the significance and the impact of the work presented in this manuscript, and for their generally positive assessment. In the revised version, we addressed the comment by adding the molecular/atomic mechanisms of action of these hybrid materials. Below, we provide response to each point, as well as a description on the changes made to address each point.

Comment: 1. *Conclusion: I am for publication of this article when some more information about the molecular/atomic mechanism of action of these materials is added.*

Response: This is a very important point, and we thank the Reviewer for bringing it to our attention. The molecular/atomic mechanism of action of these materials has been added to the text. Specifically, a new Supplementary Figure 4 was added to the Supporting Information to illustrate the changes in hydrogen bonding. Also, the following text was added on page 4, line 118 in the revised manuscript:

Added text: “The organic polymer-crystal hybrid materials were composed of two parts. PDDA/PSS//1–3 was utilized as a substrate for the hybrid materials, which are able to bend under external forces due to a combination of intermolecular interactions, and most notably, by capitalizing on the strength of hydrogen bonds (Figure 1c).^{57,60,61} The mixture of PDDA and PSS was used to improve the adhesion of the PVA/PSS/GA layer onto the crystal surface.^{62,63} PVA/PSS/GA layer functions as the driving element. As the temperature is increased or the humidity is decreased, the water molecules inside the polymer are released, causing shrinking. This change in length ultimately causes the material to bend. Conversely, when the temperature is decreased or the humidity is increased, the polymer hydroxyl groups form hydrogen bonds with the absorbed water molecules. This results in expansion and the shape is restored (Figure 2a), as qualitatively supported by the vibrational spectra (Supplementary Figure 4).^{58,64,65,66}”

New Supplementary Figure 4. Infrared spectra of polymers heated over different times with infrared lamp at 250 W (5 min, 7 min, 9 min, 11 min, and 20 min). The broad, complex band that is attributed to the O–H stretching shifts from 3452 cm^{-1} to 3399 cm^{-1} with increasing heating time, qualitatively indicating changes in the hydrogen bonding.

Reviewers' Comments:

Reviewer #1:

Remarks to the Author:

I have read the response of the authors carefully. My minor concerns have been addressed thoroughly. The response is sharp and concise. Overall, I am pleased to recommend that this work be published.

Reviewer #2:

Remarks to the Author:

The authors have modified the paper considering my recommendation and suggestions. Therefore, I recommend this paper for publication in Nat. Commun. in this current format.

Reviewer #3:

Remarks to the Author:

I have read the revised version of this ms as well as details of the Authors' answer to referees' opinions and I am happy with the changes introduced to the text of the ms. In my opinion this ms can now be accepted for publication.

Response to reviewers' comments on the manuscript "Bioinspired Soft Robots based on Organic Polymer-Crystal Hybrid Materials with Response to Temperature and Humidity"

Response to the comments from Reviewer #1:

Overall Comment: *I have read the response of the authors carefully. My minor concerns have been addressed thoroughly. The response is sharp and concise. Overall, I am pleased to recommend that this work be published.*

Response: We are delighted to know that the manuscript has been revised to meet the Reviewers' requirements, and we thank them for their constructive comments, which have led to a significant improvement in the quality of our manuscript.

.....

Response to the comments from Reviewer #2:

Overall Comment: *The authors have modified the paper considering my recommendation and suggestions. Therefore, I recommend this paper for publication in Nat. Commun. in this current format.*

Response: We are pleased that the manuscript revision has been done to the Reviewer's satisfaction. We thank the Reviewer for their constructive and encouraging comments.

.....

Response to the comments from Reviewer #3:

Overall Comment: *I have read the revised version of this ms as well as details of the Authors' answer to referees' opinions and I am happy with the changes introduced to the text of the ms. In my opinion this ms can now be accepted for publication.*

Response: We are pleased to know that the manuscript has been revised to the Reviewer's satisfaction, and we thank them for their constructive input which has helped to significantly improve the quality of our manuscript.